# Evaluation Design of EFFICHRONIC: The Chronic Disease Self-Management Programme (CDSMP) Intervention for Citizens with a Low Socioeconomic Position

**DOI:** 10.3390/ijerph16111883

**Published:** 2019-05-28

**Authors:** Siok Swan Tan, Marta M Pisano, An LD Boone, Graham Baker, Yves-Marie Pers, Alberto Pilotto, Verushka Valsecchi, Sabrina Zora, Xuxi Zhang, Irene Fierloos, Hein Raat

**Affiliations:** 1Department of Public Health, Erasmus MC University Medical Center, P.O. Box 2040, 3000 CA Rotterdam, The Netherlands; x.zhang@erasmusmc.nl (X.Z.); i.fierloos@erasmusmc.nl (I.F.); h.raat@erasmusmc.nl (H.R.); 2Fundación para el Fomento en Asturias de la Investigación Científica Aplicada y la Tecnología (FICYT), Cabo Noval St, 11, 1ºC, 33007 Oviedo–Asturias, Spain; martam.pisanogonzalez@asturias.org; 3Public Health General Directorate, Principality of Asturias (CSPA), C/ Ciriaco Miguel Vigil 9, 33006 Oviedo, Spain; anboone@ficyt.es; 4Quality Institute for Self Management Education & Training (QISMET), Harbour Court, Compass Road, North Harbour, Portsmouth, Hampshire PO6 4ST, UK; qualitygb@aol.com; 5Clinical Immunology and Osteoarticular Diseases Therapeutic Unit, Lapeyronie University Hospital, Montpellier, 371, Avenue du Doyen Gaston Giraud, 34295 Montpellier CEDEX 5, France; ympers2000@yahoo.fr (Y.-M.P.); v-valsecchi@chu-montpellier.fr (V.V.); 6Department of Geriatric Care, Orthogeriatrics and Rehabilitation, E.O. Galliera Hospital, Mura delle Cappuccine 14, 16128 Genoa, Italy; alberto.pilotto@galliera.it (A.P.); sabrina.zora@galliera.it (S.Z.)

**Keywords:** Chronic conditions, Prevention, Self-management, citizens with a low SEP, pre-post cohort study, caregivers, vulnerability

## Abstract

*Background/rationale*: The Chronic Disease Self-Management Programme (CDSMP) intervention is an evidence-based program that aims to encourage citizens with a chronic condition, as well as their caregivers, to better manage and maintain their own health. CDSMP intervention is expected to achieve greater health gains in citizens with a low socioeconomic position (SEP), because citizens with a low SEP have fewer opportunities to adhere to a healthy lifestyle, more adverse chronic conditions and a poorer overall health compared to citizens with a higher SEP. In the EFFICHRONIC project, CDSMP intervention is offered specifically to adults with a chronic condition and a low SEP, as well as to their caregivers (target population). *Study objective*: The objective of our study is to evaluate the benefits of offering CDSMP intervention to the target population. *Methods*: A total of 2500 participants (500 in each study site) are recruited to receive the CDSMP intervention. The evaluation study has a pre-post design. Data will be collected from participants before the start of the intervention (baseline) and six months later (follow up). Benefits of the intervention include self-management in healthy lifestyle, depression, sleep and fatigue, medication adherence and health-related quality of life, health literacy, communication with healthcare professionals, prevalence of perceived medical errors and satisfaction with the intervention. The study further includes a preliminary cost-effectiveness analysis with a time horizon of six months. *Conclusion*: The EFFICHRONIC project will measure the effects of the CDSMP intervention on the target population and the societal cost savings in five European settings.

## 1. Introduction

According to the World Health Organization (WHO), 35% of women and 29% of men suffer from a chronic condition, such as a cardiovascular disease, chronic respiratory disease or diabetes [1,2]. The high prevalence of chronic conditions puts a large burden on national budgets, and healthcare costs of chronic conditions reach 7% of the gross domestic product in some European countries [3,4]. Additionally, chronic conditions affect the working population by reducing their productivity and competitiveness. Fortunately—in addition to the use of medications—chronic conditions can be prevented or controlled by reducing modifiable risk factors, such as a healthy lifestyle and depression [2,5].

To achieve societal cost savings, the ability of citizens with a chronic condition to self-manage their condition for as long as possible has become increasingly important. The Chronic Disease Self-Management Programme (CDSMP) intervention was developed 40 years ago at Stanford University (the United States) and has since been implemented in more than 20 countries (e.g., Argentina, Australia, Canada, China, Costa Rica, Denmark, Spain and the United Kingdom). The evidence-based intervention was developed specifically to encourage citizens with a chronic condition and their caregivers to better manage and maintain their own health [6].

Numerous studies in different countries have demonstrated the effectiveness of the CDSMP intervention in improving self-management in key risk factors—both in general populations and disease-specific populations, such as cancer survivors [7,8] and patients with depression [9,10]. The intervention has led to improvements in physical exercise [7,9,11,12,13], depression [7,9,11,12,13], sleep and fatigue [7,9,11,12], medication adherence [7,9,14] and health-related quality of life (HR-QoL) [11,13,15]. The intervention has further shown improvements in participants’ health literacy, leading to better communication with health professionals and fewer perceived medical errors [7,11,16]. Moreover, societal benefits have been reported in terms of reduced healthcare utilization and increases in productivity, as well as reduced visits to healthcare professionals [13], visits to the hospital’s accident and emergency department room [17] and frequency and duration of hospital admissions [13]. Because the importance of lost productivity is increasingly recognized, research focus has recently shifted to employees as a target population with promising results [18]. A study carried out in the USA estimated a potential net savings of €280 per citizen and a national savings of €2.5 billion if only 5% of citizens with a chronic condition were to participate in the intervention [17].

In the existing literature on the effects of the CDSMP intervention, female, Caucasian and elderly populations are overrepresented [19]. The intervention is, however, expected to achieve greater health gains in citizens with a low socioeconomic position (SEP), because citizens with a low SEP have fewer opportunities to adhere to a healthy lifestyle, more adverse chronic conditions and a poorer overall health compared to citizens with a higher SEP [2,20]. There is scientific evidence that the SEP of citizens affects the modifiable risk factors of chronic conditions [21,22,23,24]. At the same time, citizens with a low SEP are less likely to participate in community-based interventions, training programs and research actions [19,25].

Likewise, the CDSMP intervention is expected to achieve great health gains and economic savings in caregivers, because they play a vital role in supporting citizens with a chronic condition. The association between the overall health of citizens with a chronic condition and the quality of life and perceived burden of their caregivers has been repeatedly confirmed [26]. Therefore, attention to the self-management of caregivers seems justified [27].

### 1.1. The EFFICHRONIC Project

The EFFICHRONIC project is part of the Third EU Health Programme, which addresses the chronic disease challenge (2014–2020; http://effichronic.eu/). In the project, the CDSMP intervention is offered specifically to adults with a chronic condition and a low SEP, as well as to their caregivers. Study sites in five European countries have implemented the intervention: the region of Occitanie in France, province of Genoa in Italy, region of Rotterdam in the Netherlands, principality of Asturias in Spain and the region of London in the United Kingdom.

### 1.2. Objectives

The objective of our study is to evaluate the benefits of offering the CDSMP intervention to adults with a chronic condition and a low SEP, as well as to their caregivers (the target population). The following research questions were defined:What are the effects of the CDSMP intervention for the target population in terms of self-management in healthy lifestyle, depression, sleep and fatigue, medication adherence and health-related quality of life (HR-QoL)?What are the effects of the CDSMP intervention on the health literacy, communication with healthcare professionals and prevalence of perceived medical errors?What are the societal cost savings of the CDSMP intervention in terms of reduced healthcare utilization and productivity losses among the target population?To what extent is the target population satisfied with the CDSMP intervention as a whole?

### 1.3. Study Hypotheses

We hypothesize that the intervention will provide participants with a greater ability to self-manage their chronic condition, a more healthy lifestyle, less depression, a reduction in sleeping problems and fatigue, greater medication adherence and a greater HR-QoL. We also hypothesize that the intervention will reduce the prevalence of perceived medical errors and improve communication with healthcare professionals and health literacy. We further hypothesize that society will benefit from the intervention through a reduced use of healthcare resources and a greater productivity. We hypothesize a satisfaction score of 7 or higher on a 1–10 scale for the CDSMP intervention as a whole, with higher scores representing greater satisfaction.

## 2. Materials and Methods

Trial registration: ISRCTN registry number is 70517103. Date of registration is 20/06/2018.

### 2.1. Study Design

The CDSMP intervention has been described in detail elsewhere [28,29]. In short, the intervention is based on the self-efficacy theory, and emphasizes problem solving, decision making and confidence building. It consists of a series of six tightly scripted workshops, 2.5 hours each, which are held once a week for six weeks. The intervention is offered by healthcare professionals as well as lay-persons and is attended by about 15 participants [11].

The evaluation study of the EFFICHRONIC project has a pre–post design. Data will be collected from participants before the start of the intervention (baseline) and six months later (follow up). In all study sites, every series of six workshops will preferably be led by a healthcare professional and a lay-person together. To this end, healthcare professionals and lay-persons will be recruited and trained in the CDSMP principles in the local language. The number of participants per series of workshops will be no more than 20. Written consent will be acquired from all participants.

### 2.2. Recruitment and Sample Size

A total of 2500 participants (500 in each study site) will be recruited to receive the CDSMP intervention. Recruitment of the target population will be performed in accordance with the capacity, organizational and contextual factors of each of the five study sites. In order to identify the target population, recruitment sites have been chosen intuitively based on their location in distinct environments. To be more efficient in recruiting, vulnerability maps were additionally constructed in some regions (region of Occitanie, province of Genoa and principality of Asturias), detecting EUROSTAT NUT-3 level geographical areas in which the prevalence of the target population is high. 

### 2.3. Inclusion Criteria

Citizens of at least 18 years of age with a chronic condition and a low SEP are eligible for the study, as well as their caregivers. The chronic condition is defined according to the International Classification of Primary Care (ICPC-2). Citizens are only eligible to participate when (1) their chronic condition had been present for at least six months, (2) they are able to understand the information provided in the local language and (3) they are likely to complete the duration of the study (six months).

### 2.4. Assessment of Outcomes

Data collection is done with the use of a baseline and follow up questionnaire, including a variety of outcomes. Caregivers fill out the questionnaires for themselves, not for the citizen with a chronic condition they take care of. The instruments used to measure the outcomes are described below. Instruments for which no validated translation is available are translated. Translations are discussed by the study team and adapted when needed. The baseline questionnaire will be pilot-tested in all study sites to assure its appropriateness, comprehensibility and length.

Our first research question is to evaluate the six-month effects in terms of self-management in healthy lifestyle, depression, sleep and fatigue, medication adherence and HR-QoL. Self-management is measured by six items regarding a participant’s ability to deal with fatigue, physical discomfort, emotional distress, other symptoms or health problems, different tasks and activities and other things than just taking medications (Chronic Disease Self-Efficacy (CDSE) instrument [29]). Health of lifestyle is measured via six items on physical exercise, three items on intake of fruits, vegetables and breakfast, one item on sedentary behavior (International Physical Activity Questionnaire (IPAQ) [30]), one item on smoking and one item on alcohol use (AUDIT-C [31]). Depression is measured via eight items on problems a participant may have been bothered with (Patient Health Questionnaire (PHQ-8) depression scale [32]). Sleep and fatigue are measured via two items on the severity of sleeping problems and fatigue. Medication adherence is measured via six items on medication-taking habits (Short Medication Adherence Questionnaire (SMAQ) [33]). HR-QoL is measured via 12 items on physical and mental HR-QoL (12-item short form (SF-12) [34]), five items on mobility, self-care, activity, pain and anxiety and one item on experienced current health (EQ-5D-5L [35]).

Our second research question is to evaluate the six-month effects on health literacy, communication with healthcare professionals and prevalence of perceived medical errors. Health literacy is measured via two items on information on the treatment of the chronic conditions and information provided by the healthcare professional (Health Literacy Questionnaire (HLQ) [36]). Communication with healthcare professionals is measured via three items on preparing a list of questions, asking questions and discussing personal problems [29]. Perceived medical errors are measured via three items on the understandability of a healthcare professional’s explanation on things and prevalence of perceived medical errors (American Association of Retired Persons (AARP) ‘survey beyond 50.09’ questionnaire).

The third research question regards societal cost savings in terms of reduced healthcare utilization and productivity losses. Healthcare utilization is measured via four items on frequency of visits to healthcare professionals, frequency of visits to a hospital’s accident and emergency department and frequency and duration of hospital admissions (SMRC Health Care Utilization questionnaire [29]). Productivity losses are measured via six items on lost productivity at paid work and three items on lost productivity at unpaid work (Productivity Costs Questionnaire (PCQ) [37].

The baseline questionnaire additionally includes socio-demographics (age, gender, country of birth, educational level and employment situation), whereas the follow up questionnaire includes three items on experienced improvement in the most important outcomes of the CDSMP intervention (problem solving, decision making and confidence building), one item on confidence in the national health system, one item on improvement experienced in interpersonal communication skills and one item on satisfaction with the intervention as a whole.

### 2.5. Stratification

The EFFICHRONIC project supports the validation of the selfy-MPI, a newly developed instrument to stratify vulnerable citizens. It is an adjusted version of the Multidimensional Prognostic Index (MPI), which was originally developed to define the severity grade of mortality risk in geriatric patients with chronic conditions [38]. In contrast with the MPI, the selfy-MPI may be used in a younger population, allows for self-report and incorporates SEP as one of its dimensions.

The selfy-MPI comprises eight dimensions. The first three dimensions measure the participant’s level of independence using 18 items on independence of (instrumental) activities in daily living ((I)ADL) [39,40]. The fourth dimension concerns a Test Your Memory (TYM) test consisting of 10 tasks, including the ability to copy a sentence, knowledge of words and their meanings and recall ability [41]. The fifth dimension measures nutritional aspects by means of the Mini Nutritional Assessment (MNA) [42]. The sixth dimension measures comorbidity using the Cumulative Illness Rating Scale (CIRS) [43]. The seventh dimension measures the use of medications via two items on the intake and variety of medications. Finally, an adapted version of Gijón’s social-familial evaluation scale (SFES) [44] assesses the participant’s household composition, net monthly household income, housing situation, social relationships and social support.

The selfy-MPI is validated by using it to retrospectively stratify our study population. It may be used as a prospective stratification instrument in the future.

### 2.6. Power Considerations

In a study on the effects of the CDSMP intervention, the sub-score ‘well-being’ and ‘energy’ of the SF-36 improved with 6.0 +/– 19.1 and 4.3 +/– 23.2, respectively after six months [45]. The lowest Cohen’s d is therefore 0.185 for physical and mental HR-QoL. In our study, we wish to demonstrate a statistically significant difference on the two subscales of the SF-12 in the five study sites. Thus, the principal conclusion of the study will rely on 10 tests. Therefore, we apply Bonferroni’s correction on the significance threshold and consider an alpha risk of 0.005 for the tests of the objective of our study. To show a statistically significant effect size of 0.185—with a bilateral alpha risk of 0.005, a power of 0.9 and a correlation between baseline and follow up measurements of 0 via a paired Student’s t test—we need to analyze 978 subjects. Assuming a drop-out rate of 20%, we need to include 1223 subjects. The study protocol will include 2500 subjects (500 per study site). Assuming a 20% loss to follow-up between baseline and follow up, we expect to receive complete data of 2000 participants at follow up, which will be sufficient for analysis of the research questions.

### 2.7. Data Management and Statistical Analysis

The Erasmus MC University Medical Center is responsible for the data-management, analysis and reporting. All data is handled confidentially and scientific data is stored anonymously.

In addition to descriptive statistics, tests for normal distribution of the outcomes and socio-demographics will be performed using a Kolmogorov–Smirnov test. Differences between baseline and follow up measurements will be assessed by means of the paired t test (for variables showing a normal distribution), Mann–Whitney U test (for variables not normally distributed) or Pearson’s chi-square test (for variable fractions). To determine the relationship of an outcome (at follow up) with explanatory variables, ordinary least squares (OLS) regression will be performed for continuous outcome variables with the outcome (at baseline) and (change in) other outcomes and socio-demographics as explanatory variables. Logistic regression will be performed for dichotomous outcome variables. In addition, the above-mentioned analyses will be repeated for each study site separately, for different risk groups as identified by the selfy-MPI and possibly for other subgroups based on variables that are likely to influence the effect of the intervention (e.g., age, sex).

Using the baseline measurement as the control group, a preliminary cost-effectiveness analysis will be performed from a societal and healthcare perspective with a time horizon of six months. Healthcare costs for individual participants will be determined by multiplying resource use (visits to healthcare professionals, visits to the hospital’s accident and emergency department and hospital admissions) with corresponding unit prices for 2019. Productivity losses for individual participants will follow from the PCQ. Utility values will be obtained through the EQ-5D-5L instrument. 

Finally, a budget impact analysis (BIA) will estimate the financial consequences of implementing the intervention. BIA is very valuable in supporting the adoption of policies for the prevention of chronic conditions because it takes accessibility and affordability constraints into account when considering cost-effectiveness. To estimate the societal cost savings of the CDSMP intervention, the mean societal costs per participant will be extrapolated to the prevalence of the target population in the countries of the study sites.

### 2.8. Dissemination

The scientific project results will be published in peer-reviewed journals and disseminated at (inter)national conferences. To further disseminate the knowledge to all stakeholders we use the project website (http://effichronic.eu/) and social media. Additionally, quality control of the project is performed by an advisory board including experts from a representative number of European countries and by the University of Valencia (UVEG).

## 3. Discussion

This evaluation study aims to evaluate the benefits of offering the CDSMP intervention to adults with a chronic condition and low SEP, as well as to their caregivers. This study has several strengths. To our knowledge this is one of the first European studies that implements an evidence-based intervention specifically in citizens with a low SEP. Citizens with a low SEP are an understudied population with fewer opportunities to adhere to a healthy lifestyle, more adverse chronic conditions and a poorer overall health [2,20]. Recruiting the target population in different settings will provide information on the generalizability of the approach in various European settings. This could generate a wider acceptance of the CDSMP intervention and facilitate implementation of other health surveys, interventions and health programs in this population.

Furthermore, cost-effectiveness data regarding self-management programs are limited. The EFFICHRONIC project will provide insight into potential societal cost savings resulting from a reduction in healthcare utilization and productivity losses.

The study has some limitations, and we expect to encounter various challenges. Firstly, the recruitment of the target population will be performed in accordance with the capacity, organizational and contextual factors of each of the five study sites. Although this implies that the recruitment will be different in each study site, evaluating the effectiveness of approaches in different (international) settings is valuable. Secondly, we decided that it is not desirable to include a control group. To avoid feelings of exclusion, the intervention will be available to all citizens meeting our inclusion criteria. To conduct the cost-effectiveness analysis, the baseline measurement will be used as the ‘control value’. Thirdly, we will have to trade-off the length of our questionnaire with the difficulties citizens with a low SEP may have in answering the questions. We will try to capture the most important risk factors; however, it is likely that we will miss relevant variables.

## 4. Conclusions

As the high prevalence of chronic conditions poses a challenge for Europe, new ways of achieving sustainable health systems are necessary. Increasing the ability of citizens with a chronic condition to take care of themselves for as long as possible may provide better outcomes. The EFFICHRONIC project will further elucidate whether such an approach could be feasible and (cost-) effective for populations with low SEPs, as well as to their caregivers, in different settings.

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
