# Peer review of "Evaluation Design of EFFICHRONIC: The Chronic Disease Self-Management Programme (CDSMP) Intervention for Citizens with a Low Socioeconomic Position"

_ijerph, 2019, doi:10.3390/ijerph16111883_

Round 1

Reviewer 1 Report

A very well written article, on an important topic and public health intervention. My primary concern is related to the structure of the article. The articles builds up to an analysis that hasn't happened yet. The structure reads more as a research proposal than a research article. I was left wondering what the scholarly contribution of this paper is (given that you are discussing research which has not occurred yet and therefore do not have outcomes to share). 

To correct this concern, I would consider the following. 

Reorient the paper as a literature review. Expand and deepen the literature review on the proposed intervention. The existing literature review would be sufficient for a research paper, but given this paper does not have any results, the scholarly contribution can be the depth of your literature review. 

Move your proposed research discussion into the discussion of the article. Frame this as building upon the insights you have derived from your review of the literature. I would not add more content to this research proposal (which is very well written), but would re frame this more clearly as being directly influenced by your review of the relevant literature. 

You have a well written article and interesting topic. My suggestion is to re frame the article so to better represent its scholarly contribution. 

Author Response

Reviewer 1: A very well written article, on an important topic and public health intervention. My primary concern is related to the structure of the article. The articles builds up to an analysis that hasn't happened yet. The structure reads more as a research proposal than a research article. I was left wondering what the scholarly contribution of this paper is (given that you are discussing research which has not occurred yet and therefore do not have outcomes to share). Our manuscript indeed is a study protocol (not a research article) in which no results are described. To correct this concern, I would consider the following. Reorient the paper as a literature review. Expand and deepen the literature review on the proposed intervention. The existing literature review would be sufficient for a research paper, but given this paper does not have any results, the scholarly contribution can be the depth of your literature review. Move your proposed research discussion into the discussion of the article. Frame this as building upon the insights you have derived from your review of the literature. I would not add more content to this research proposal (which is very well written), but would re frame this more clearly as being directly influenced by your review of the relevant literature. You have a well written article and interesting topic. My suggestion is to re frame the article so to better represent its scholarly contribution. In the introduction we have now provided an expanded and deepened literature review on the CDSMP intervention: ‘Numerous studies in different countries have demonstrated the effectiveness of the CDSMP intervention in improving self-management in key risk factors – both in general populations and disease specific populations, such as cancer survivors (7, 8) and patients with depression (9, 10). The intervention led to improvements in physical exercise (7, 9, 11-13), depression (7, 9, 11-13), sleep and fatigue (7, 9, 11, 12), medication adherence (7, 9, 14) and health-related quality of life (HR-QoL)(11, 13, 15). The intervention further showed improvements in participants’ health literacy, leading to better communication with health professionals and fewer perceived medical errors (7, 11, 16). Moreover, the reported improvements significantly reduced healthcare utilization and productivity losses. Significant decreases were reported in visits to healthcare professionals (13), visits to the hospital’s Accident and emergency department room (17) and in the frequency and duration of hospital admissions (13). Because the importance of lost productivity at both paid and unpaid work is increasingly recognized, research focus recently shifted to employees as a target population with promising results (18). A study carried out in the USA estimated the potential net savings to be €280 per citizen and national savings to be €2.5 billion if only 5% of citizens with a chronic condition would participate in the intervention (17). In the existing literature on the effectiveness of the CDSMP intervention, female, Caucasian and elderly populations are overrepresented (19). The intervention is, however, expected to achieve greater health gains in citizens with a low socioeconomic position (SEP), because citizens with a low SEP have fewer opportunities to adhere to a healthy lifestyle, more adverse chronic conditions and a poorer overall health compared to citizens with a higher SEP (2, 20). There is scientific evidence that the SEP of citizens affects the modifiable risk factors of chronic conditions (21-24). At the same time, citizens with a low SEP are less likely to participate in community-based interventions, training programmes and research actions (19, 25). Likewise, the CDSMP intervention is expected to achieve health gains and economic savings in caregivers, because they play a vital role in supporting citizens with a chronic condition. The association between the overall health of citizens with a chronic condition and the quality of life and perceived burden of their caregivers has been repeatedly confirmed (26). Therefore, attention to the self-management of caregivers seems justified (27).’ (line 76-103)

Reviewer 2 Report

The authors have presented a protocol for a study, EFFICHRONIC, that aims to evaluate the Chronic Disease Self-Management Programme (CDSMP). There are several concerns with this protocol that are presented below for authors' consideration.

1. The grammar and English are very poor. Authors should consult the services of an English editor for improving the quality of the manuscript.

2. The title is ambiguous and makes no sense to the reader. This should be revised.

3. The abstract is meant to be a summary of the manuscript with clearly written background/rationale, study objectives, methods, results, and conclusion. The current abstract mentions two interventions, CDSMP and EFFICHRONIC, and the study objectives are ambiguous and do not appear feasible. The sentence in lines 46-49 is incomplete. In line 41, what is meant by "their disease progression is less favorable?" 

4. Introduction: Line 72: "What is meant by "..and maintain their health status?"

The study objectives are confusing: 

"EFFICHRONIC intends to empower the target population to self-manage their chronic conditions through an integrated approach, gathering citizens suffering from a variety of chronic conditions and evaluating (cost-) effectiveness." 

"The main objective of the evaluation study is to appraise the CDSMP intervention in terms of benefits for the target population."

5. Methods: What is the reason for including caregivers with low SEP in the study? How will low SEP be determined during the recruitment process? What is the role of the caregivers based on the study objective (lines 174-179) "Our primary objective is to evaluate the 6-month improvement of self-management in healthy life style, depression, sleep and fatigue, adherence to medications and HR-QoL. Self-management is measured with six items on someone’s ability to deal with fatigue, physical discomfort, emotional distress, other symptoms or health problems, different  tasks and activities and other things than just taking medications (Chronic Disease Self-Efficacy 178 instrument; CDSE-6)."

In line 190, "A secondary set of outcome measures concern the prevalence of experienced medical errors, communication with healthcare providers and health literacy." The basis for assessing the prevalence of medical errors is not clear. Since all of the study participants do not receive healthcare from the same provider, this outcome measure will have high variation, and may not be useful information in improving health service performance, based on your study design. How does CDSMP impact medical errors? 

The authors need to do a thorough revision of this protocol in order for the study to provide meaningful contribution to scientific literature.

Author Response

Reviewer 2: The authors have presented a protocol for a study, EFFICHRONIC, that aims to evaluate the Chronic Disease Self-Management Programme (CDSMP). There are several concerns with this protocol that are presented below for authors' consideration. 1. The grammar and English are very poor. Authors should consult the services of an English editor for improving the quality of the manuscript. We have revised the English, in particular in the introduction and discussion of the manuscript. We could still consult an English corrector if the editor wishes so. 2. The title is ambiguous and makes no sense to the reader. This should be revised. The title was changed into: ‘Evaluation design of EFFICHRONIC: the Chronic Disease Self-Management Programme (CDSMP)-intervention in citizens with a low socioeconomic position’ (line 2-5) 3. The abstract is meant to be a summary of the manuscript with clearly written background/rationale, study objectives, methods, results, and conclusion. The abstract now contains headers for background/rationale, study objective, methods and conclusion. The content was improved: ‘Background/rationale: The Chronic Disease Self-Management Programme (CDSMP)-intervention is an evidence-based programme which aims to encourage citizens with a chronic condition and their caregiver to better manage and maintain their own health. The CDSMP intervention is expected to achieve greater health gains in citizens with a low socioeconomic position (SEP), because citizens with a low SEP have fewer opportunities to adhere to a healthy lifestyle, more adverse chronic conditions and a poorer overall health compared to citizens with a higher SEP. In the EFFICHRONIC project, the CDSMP intervention is offered specifically to adults with a chronic condition and with a low SEP and their caregivers (target population). Study objective: The objective of our study is to evaluate the benefits of offering the CDSMP intervention to the target population. Methods: A total of 2,500 participants (500 in each country) are recruited to receive the CDSMP intervention. The evaluation study has a pre-post design. Data will be collected from participants before the start of the intervention (baseline) and 6 months later (follow up). Benefits of the intervention include self-management in healthy lifestyle, depression, sleep and fatigue, medication adherence, health-related quality of life, prevalence of experienced medical errors, communication with healthcare professionals, health literacy and satisfaction with the intervention. The study further includes a preliminary cost-effectiveness analysis with a time horizon of 6 months. Conclusion: The EFFICHRONIC project will measure the effects of the CDSMP intervention on the target population and the societal cost savings in five European settings.’ (line 38-55) The current abstract mentions two interventions, CDSMP and EFFICHRONIC, We agree with the reviewer that referring to both CDSMP and EFFICHRONIC as being the intervention is confusing. We now refer to the CDSMP intervention and EFFICHRONIC project throughout the manuscript. and the study objectives are ambiguous and do not appear feasible. In the previous version of our manuscript we stated the objective of the CDSMP intervention, the objective of the EFFICHRONIC project and the objective of the evaluation study. We agree with the reviewer that this is confusing. In the revised version, we only state the objective of the evaluation study, as follows: ‘The objective of our study is to evaluate the benefits of offering the CDSMP intervention to adults with a chronic condition and with a low SEP and their caregivers (the target population).’ (line 45-47) The sentence in lines 46-49 is incomplete. We completed the sentence as follows: ‘Benefits of the intervention include self-management in healthy lifestyle, depression, sleep and fatigue, medication adherence, health-related quality of life, health literacy, communication with healthcare professionals, prevalence of perceived medical errors and satisfaction with the intervention.’ (line 49-52) In line 41, what is meant by "their disease progression is less favorable?" The phrase was changed in the abstract as well as in the introduction into: ‘…, because citizens with a low SEP have fewer opportunities to adhere to a healthy lifestyle, more adverse chronic conditions and a poorer overall health compared to citizens with a higher SEP’ (line 41-43 and 93-95) 4. Introduction: Line 72: "What is meant by "..and maintain their health status?" The phrase was changed in the abstract as well as in the introduction into: ‘… to encourage citizens and their caregiver with a chronic condition to better manage and maintain their own health’ (line 39-40 and 74-75) The study objectives are confusing: "EFFICHRONIC intends to empower the target population to self-manage their chronic conditions through an integrated approach, gathering citizens suffering from a variety of chronic conditions and evaluating (cost-) effectiveness." "The main objective of the evaluation study is to appraise the CDSMP intervention in terms of benefits for the target population." In the previous version of our manuscript we stated the objective of the CDSMP intervention, the objective of the EFFICHRONIC project and the objective of the evaluation study. We agree with the reviewer that this is confusing. In the revised version, we only state the objective of the evaluation study, as follows: ‘The objective of our study is to evaluate the benefits of offering the CDSMP intervention to adults with a chronic condition and with a low SEP and their caregivers (the target population).’ (line 45-47) 5. Methods: What is the reason for including caregivers with low SEP in the study? In the introduction we now explain the reason for including caregivers: ‘Likewise, the CDSMP intervention is expected to achieve health gains and economic savings in caregivers, because they play a vital role in supporting citizens with a chronic condition. The association between the overall health of citizens with a chronic condition and the quality of life and perceived burden of their caregivers has been repeatedly confirmed (26). Therefore, attention to the self-management of caregivers seems justified (27).’ (line 99-103) How will low SEP be determined during the recruitment process? In the methods-section we explain: ‘Recruitment of the target population is performed in accordance with the capacity, organizational and contextual factors of each of the five study sites. In order to identify the target population, recruitment sites are chosen intuitively based on their location in distinct environments. To be more efficient in recruiting, vulnerability maps were additionally constructed in some regions (region of Occitanie, province of Genoa and principality of Asturias), detecting EUROSTAT NUT-3 level geographical areas in which the prevalence of the target population is high.’ (line 148-153) What is the role of the caregivers based on the study objective (lines 174-179) "Our primary objective is to evaluate the 6-month improvement of self-management in healthy life style, depression, sleep and fatigue, adherence to medications and HR-QoL. Self-management is measured with six items on someone’s ability to deal with fatigue, physical discomfort, emotional distress, other symptoms or health problems, different tasks and activities and other things than just taking medications (Chronic Disease Self-Efficacy 178 instrument; CDSE-6)." We have clarified that: ‘Caregivers fill out the questionnaires for themselves; not for the citizen with a chronic condition they take care of.’ (line 163-164) In line 190, "A secondary set of outcome measures concern the prevalence of experienced medical errors, communication with healthcare providers and health literacy." The basis for assessing the prevalence of medical errors is not clear. Since all of the study participants do not receive healthcare from the same provider, this outcome measure will have high variation, and may not be useful information in improving health service performance, based on your study design. How does CDSMP impact medical errors? In the introduction we have included: ‘The intervention further showed improvements in participants’ health literacy, leading to better communication with health professionals and fewer perceived medical errors (7, 11, 16).’ (line 81-82) The authors need to do a thorough revision of this protocol in order for the study to provide meaningful contribution to scientific literature. We have revised the protocol based on earlier points (see above).

Reviewer 3 Report

Lines 10,21, 26,28,30,32 - I think the authors write department with ''D''

Please include in Introduction the aim of this project and also, which is the novelty of the research.

Please detailed the instruments measures....see lines 174-188....the presentation of this subsection is too simple and unclear....please detailed

Author Response

Reviewer 3: Lines 10,21, 26,28,30,32 - I think the authors write department with ''D'' We have changed ‘department’ into ‘Department’ (line 9-31) Please include in Introduction the aim of this project and also, which is the novelty of the research. In the introduction we now emphasize that : ‘In the existing literature on the effectiveness of the CDSMP intervention, female, Caucasian and elderly populations are overrepresented (19). The intervention is, however, expected to achieve greater health gains in citizens with a low SEP, […] Likewise, the CDSMP intervention is expected to achieve health gains and economic savings in caregivers […].The objective of our study is to evaluate the benefits of offering the CDSMP intervention to adults with a chronic condition and with a low SEP and their caregivers (the target population). Please detailed the instruments measures....see lines 174-188....the presentation of this subsection is too simple and unclear....please detailed In line with the rest of the section, we have revised the subsection as follows: ‘Our first research question is to evaluate the 6-month effects in terms of self-management in healthy lifestyle, depression, sleep and fatigue, medication adherence and HR-QoL. Self-management is measured with six items on someone’s ability to deal with fatigue, physical discomfort, emotional distress, other symptoms or health problems, different tasks and activities and other things than just taking medications (Chronic Disease Self-Efficacy instrument; CDSE-6 (29)). Healthy life style is measured with six items on physical exercise, three items on intake of fruits, vegetables and breakfast, one item on sedentary behavior (International Physical Activity Questionnaire; IPAQ (30)), one item on smoking and one item on alcohol use (AUDIT-C (31)). Depression is measured with 8 items on problems someone may have been bothered with (Patient Health Questionnaire depression scale; PHQ-8 (32)). Sleep and fatigue are measured with two items on the severity of sleeping problems and fatigue. Medication adherence is measured with six items on medication-taking habits (Short Medication Adherence Questionnaire; SMAQ) (33)). HR-QoL is measured with 12 items on physical and mental HR-QoL (12-item short-form; SF-12 (34)), five items on mobility, self-care, activity, pain and anxiety as well as one item on experienced current health (EQ-5D-5L (35)).’ (line 169-183)

Round 2

Reviewer 2 Report

The authors have adequately addressed my concerns from the initial version of the manuscript, and I believe this current version is much improved.